# Effects of Climate on Douglas-fir (*Pseudotsuga menziesii* (Mirb.) Franco) Growth Southeast of the European Alps

**DOI:** 10.3390/plants11121571

**Published:** 2022-06-14

**Authors:** Tom Levanič, Hana Štraus

**Affiliations:** 1Department of Forest Yield and Silviculture, Slovenian Forestry Institute, Večna Pot 2, SI-1000 Ljubljana, Slovenia; 2Faculty of Mathematics, Natural Sciences and Information Technologies, University of Primorska, Glagoljaška 8, SI-6000 Koper, Slovenia; 3Department of Forestry and Renewable Forest Resources, Biotechnical Faculty, University of Ljubljana, Večna Pot 83, SI-1000 Ljubljana, Slovenia; hana.straus@bf.uni-lj.si

**Keywords:** climate change, climate response, drought, radial increment, dendrochronology

## Abstract

Douglas-fir (*Pseudotsuga menziesii* (Mirb.) Franco) is a non-native tree species in Slovenia with the potential to partially replace Norway spruce in our native forests. Compared to spruce, it has several advantages in terms of volume growth, wood quality and tolerance to drought. This is important given the changing climate in which spruce is confronted with serious problems caused by increasing temperatures and drought stress. At three sites (one on non-carbonate bedrock and deep soils, and two on limestone with soil layers of varying depths), 20 Douglas-fir and 20 spruce per site were sampled in order to compare their radial growth response to climate and drought events. The radial growth of Douglas-fir exceeds that of spruce by about 20% on comparable sites. It is more responsive to climate than spruce. Above-average temperatures in February and March have a significant positive effect on the radial growth of Douglas-fir. In recent decades, above-average summer precipitation has also had a positive influence on the radial growth of Douglas-fir. Compared to spruce, Douglas-fir is less sensitive to extreme drought events. Our results indicate that Douglas-fir may be a good substitute for spruce in semi-natural managed forest stands in Slovenia. The planting of Douglas-fir should be allowed in Slovenian forests, but the proportion of it in forest stands should be kept lower than is the case with spruce today.

## 1. Introduction

Climate change is having a major impact on the world’s ecosystems. Owing to the longevity of trees, forest ecosystems cannot quickly adapt to changes in the environment. In Europe, climate change is mainly reflected in increasing temperatures (0.85 °C increase from 1880 to 2012) [1], more frequent droughts and more extreme weather events [2]. As a result, higher mortality of individual tree species has already been observed in some forest ecosystems due to drought stress and related factors [3]. Norway spruce (*Picea abies* (L.) Karst), one of the most economically important tree species in Europe, has been the most affected. Slovenia has also experienced massive spruce dieback in recent years due to climate change, natural disasters and the spread of bark beetles. Empirically-based random forest models show that the area of potential spruce sites in Slovenia will decrease in the future [4], and that deciduous forests will slowly replace coniferous forests [5]. As the proportion of spruce in the growing stock decreases, the economic benefits of the forest are expected to decrease, and significant changes in the wood processing industry are also expected. In order to maintain the socio-economic functions of forests, it is necessary to find, among other things, tree species that possess wood properties similar to those of spruce, and that are more resistant and resilient to extreme climatic conditions and drought. An important limiting factor for the future growth of species will most likely be a lack of accessible water [3].

One way to mitigate the negative consequences of spruce dieback is by introducing other species, such as Douglas-fir (*Pseudotsuga menziesii* (Mirb.) Franco), to close-to-nature forests as a partial replacement for Norway spruce [6]. Due to its fast growth and high-quality wood, Douglas-fir is the most common non-native tree species in many European countries, including Slovenia. Currently, the planting of Douglas-fir is prohibited in Slovenia (Official Gazette of the Republic of Slovenia, No. 96/04), although it was planted in the past and today accounts for 0.5% of the growing stock [7].

The natural distribution range of Douglas-fir extends along the entire Pacific coast of North America, from British Columbia to Mexico. It thrives in a wide range of climatic conditions, but proliferates best in habitats with high humidity and deep, aerated soils [8]. In Slovenia, as in the rest of the central Europe, it grows best at altitudes ranging from 500 to 1000 m above sea level in the belt of beech and silver fir–beech forests [9]). It requires high humidity and a moist, well-aerated and deep soil for rapid growth. It grows poorly in shallow limestone soils and in compacted soils with standing water [10].

Douglas-fir originates from an environment with warm, dry summers, and therefore is physiologically better adapted to a lack of water in the soil compared to Norway spruce. Small leaf stomata, heavily waxed needles and an effective stomata regulating mechanism allow Douglas-fir to use water sparingly, and maintain a positive photosynthetic ratio [11]. Compared to Norway spruce, the root system of Douglas-fir is deeper, which makes it more resistant to drought and windthrow [12]. Resistance to different climatic factors largely depends on the variety and provenance of the species [13]. Despite these physiological adaptations, extreme droughts in recent years have also resulted in needle shed, crown reduction and mortality of Douglas-fir trees in France [14]. Various sources have also noted that the growth of Douglas-fir has also been negatively affected, especially in the late winter months, as it is sensitive to frost [15,16,17].

Previous studies have shown that the impact of Douglas-fir on European forest ecosystems is comparable to that of native conifers [18], and is less than that of other non-native species [19]. It has the greatest impact when planted in monocultures over large areas, making plantations particularly problematic. Its impact on the soil is also small and difficult to determine due to the complexity of the various factors affecting soil chemistry. The root system is deeper than that of spruce, which reduces competition for nutrients in the rhizosphere. It also appears to be more resistant to drought and the negative effects of climate change in general than spruce. Under suitable conditions, it is potentially invasive in some parts of Europe [20,21], but studies indicate that its invasive potential is not likely to increase due to climate change, as the shallow root system of young trees is vulnerable to drought conditions [19].

Currently, Douglas-fir is not troubled by many pests in Europe, although in its natural habitats in North America, it is affected by a wide variety of pests, including some very damaging and destructive insects (e.g., Douglas-fir tussock moth (*Orgyia pseudotsugata* McDunnough), Douglas-fir beetle (*Dendroctonus pseudotsugae*, Hopkins) and Western spruce bud worm (*Choristoneura occidentalis* Freeman)) [22].

Little is known about the growth of Douglas-fir and its response to climate in Slovenia. The aim of the study was therefore to (1) investigate the radial growth of Douglas-fir on sites with different productivity; (2) analyse the climatic factors that significantly influence the growth of Douglas-fir in order to identify potentially critical meteorological factors for its radial growth and growth in general; (3) compare the response of Douglas-fir and spruce to climatic factors on sites with different productivity; (4) analyse the response of Douglas-fir in years with markedly dry and hot summers and compare it to the response of Norway spruce; and (5) determine whether Douglas-fir can be a good substitute for spruce, and whether replacement of Norway spruce is necessary and appropriate due to spruce’s declining vitality caused by climate change.

## 2. Results

### 2.1. Radial Growth of Douglas-Fir and Norway Spruce on Three Sites

The radial growth of Douglas-fir and Norway spruce was studied on three different locations with different bedrock and soil depths. Basic information about age, diameter at breast height (DBH) and average width of the annual increment is presented in Table 1. Douglas-fir trees on the studied sites generally have a larger DBH than Norway spruce trees, with radial increments similarly higher in Douglas-fir than in Norway spruce (Table 1).

Douglas-fir on site CE, which is characterised by non-carbonate bedrock and deep soil, clearly outperforms Norway spruce. On this site, both tree species are growing in the same forest stand with the same bedrock and soil depth. Radial growth was comparable in the period 1940–1990, but in the period 1990–2020 the radial growth of Douglas-fir on site CE was significantly greater than that of Norway spruce. Moreover, while the radial growth of Norway spruce declined after 1998, Douglas-fir’s radial increment slowly increased (Figure 1).

On limestone bedrock and at variable soil depths of the Dinaric Karst, Douglas-fir exhibits variable performance in comparison to spruce. In deeper soils on site PO-1, there is very little difference between the radial growth of Douglas-fir and Norway spruce; both tree species grow more or less similarly and respond to environmental factors in the same way. There is a slight difference in radial growth in favour of Douglas-fir, but it is not statistically significant. The radial growth of Douglas-fir has been decreasing in the last decade and is equal to that of Norway spruce (Figure 1).

On site PO-2 (also Dinaric Karst) with shallow soil and limestone bedrock, the radial growth of Douglas-fir is higher than that of Norway spruce, regardless of the fact that the spruce trees there are older than the Douglas-fir trees. Both species show a similar response to environmental factors. We observed a decrease in the radial growth of Douglas-fir after 1998, with the same decrease also visible in Norway spruce. A recovery in the radial growth of Douglas-fir was observed after 2013. A similar recovery is also visible in Norway spruce, but it is much smaller in terms of absolute numbers (Figure 1).

### 2.2. Climate–Growth Relationship of Douglas-fir

The general growth response of Douglas-fir and Norway spruce to climate differs significantly (Figure 2), with Douglas-fir having a much clearer radial growth response to climate than Norway spruce. The above-average radial growth of Douglas-fir is significantly associated with above-average winter temperatures, particularly in February and March, while that of Norway spruce is correlated with above-average precipitation in the summer months, specifically June and July. Above-averages temperature in June and July have a negative impact on the radial growth of Norway spruce and no statistically significant impact on that of Douglas-fir.

In contrast to the general response of trees to climatic factors in the growing season, Douglas-fir is particularly sensitive to above-average temperatures in February and March. Higher than average temperatures in February and March have a positive influence on radial growth for the growing season that follows. This influence on radial growth was observed on all studied sites; however, on PO-1 this relationship was visible but not significant. The highest correlations between radial growth and February–March temperatures were found on the most productive site CE, followed by PO-2 and PO-1 (Figure 2, left panel).

On the least productive and drier site PO-2, precipitation in the growing season also plays important role, especially the lack of precipitation in July, which is indicated by a high positive correlation. A similar response was also observed on site CE, where the correlation is smaller but still significant. On site PO-1, we found a low but still significant negative correlation with above-average precipitation in the previous October. As the interpretation of such a relationship is difficult, we believe that it is just an artefact of the bootstrapped calculation of the correlation coefficient. Additionally, on site PO-2 we found a significant positive effect of precipitation on the radial growth of Douglas-fir in April and a negative effect on it in May (Figure 2, left panel). It seems that above-average precipitation in April has a beneficial influence on the radial growth of Douglas-fir, while above-average temperature in May has a negative influence on radial growth of Norway spruce. It is quite possible that above-average precipitation in May relates to cold air intrusion, which often brings very low temperatures and even snow in the first half of May.

The dependence of spruce on February and March temperatures is significant only on the most productive site CE, while on sites PO-1 and PO-2, we found a significant response of radial growth to climate only in the summer months. Specifically, on the productive PO-1 site, above-average temperature in July has a negative influence on radial growth, while on the least productive site PO-2, above-average temperatures in June, July and August have a significant negative impact on radial growth (Figure 2, right panel).

Precipitation plays a more important role in the radial growth of Norway spruce than temperature. Above-average precipitation in June and July has a positive effect on tree-ring width on PO-1 and PO-2 sites, while on site CE only above-average precipitation in July has a positive influence on tree-ring width (Figure 2, right panel).

The combination of the positive impact of precipitation and negative impact of temperature means that the radial growth of Norway spruce on the least productive site PO-2 and, to some degree, also on the more productive sites CE and PO-1, is negatively affected by a hot and dry climate in the summer months (Figure 2, right panel).

Both studied tree species have a radically different response to climate. For Douglas-fir, February and March are important months for radial growth in the subsequent growth period. If February and March temperatures are above average, we can expect higher radial growth, and vice versa. Precipitation has less influence on the radial growth of Douglas-fir, most likely as a result of its deep root system. Compared to Douglas-fir, Norway spruce is not as sensitive to winter temperatures. However, above-average precipitation in June and July and above-average temperature in July have a positive and negative influence, respectively. As a result of its shallow root system, Norway spruce is more susceptible to hot and dry summers than Douglas-fir, regardless of site productivity.

Analysis of the radial growth response to climate using bootstrapped moving window correlation provides insight into the temporal response of the studied trees to climate, and into the response of trees to changing climatic conditions over time. This is particularly important given that some climatic factors might have become more significant and some less significant over time. Additionally, by means of temporal analysis with moving window correlation (Appendix A and Appendix B), it is also possible to determine whether the radial growth response differs from the general tree-ring width response to climate (as in Figure 2), and whether responses become stable over time or not.

There are important differences between Douglas-fir and Norway spruce with respect to temporal response. The temporal response of Douglas-fir to temperature on all studied sites is clear, statistically significant and stable over time. Above-average temperatures in February and March are important drivers of Douglas-fir growth: warmer late winter months are associated with higher radial increments in the summer months.

The temporal stability of the precipitation signal in Douglas-fir is less pronounced. Only precipitation in July seems to play a role in the higher radial increment on the least productive and most drought-exposed site PO-2, which is expected as this site was selected because of its location on a ridge and its shallow soil. On sites CE and PO-1, precipitation plays some role in July, but is less important and not statistically significant (see Appendix A).

The temporal response of Norway spruce is less pronounced than that of Douglas-fir. On the most productive site CE, above-average March temperature has had an influence on radial growth in the last three decades, and July precipitation has had an influence on radial growth in the last two decades. We also observed that the climate–growth relationship has not been stable over time, and that statistical significance is achieved only occasionally.

On the productive site PO-1 (limestone bedrock on deep soil), the influence of July precipitation on the radial growth of Norway spruce has been stable over time, and its importance has been increasing. June precipitation has also gained importance in the last three decades. The influence of temperature on the radial growth of spruce on site PO-1 is statistically significant in July, but the values are not very high.

On the least productive site PO-2 (limestone bedrock, shallow soil), precipitation in July played an important role throughout the entire study period. The temporal stability of the signal is well defined and moving correlations are high. We also observed a temporally stable negative influence of above-average temperature in July. This means that Norway spruce trees on site PO-2 are negatively affected by the occurrence of drought, and that drought is becoming more pronounced (see Appendix B).

### 2.3. Douglas-Fir Response in Extreme Years

The response in extreme years, whether they were extremely hot and dry or cold and wet, indicates the ability or adaptability of a tree species to cope with climatic extremes. Even if we do not find statistically significant correlations with climate data for a tree species, which is quite common in the case of lowland oaks [23,24], this does not mean that a tree species does not respond to climate. When we record a weak climatic response of a tree species, or when we are interested in how a tree species could respond through adjustments in the radial increment in harsher climate conditions, it is useful to consider the response in extreme climatic conditions (e.g., hot and dry summers). We analysed pointer years for both tree species on all three sites; however, in this paper we present only the analysis of pointer years for Douglas-fir, and only basic comparisons with respect to the Norway spruce response.

Douglas-fir has fewer negative pointer years compared to spruce, so it is slightly more tolerant to a lack of precipitation in summer, probably due to its deeper root system. From typical dry and hot years, such as 1976, 1980, 1992, 2003, 2013 and 2015, which were detected in spruce at all studied sites, only one negative pointer year (2003) was detected in Douglas-fir on the PO-1 and PO-2 sites. On the other hand, we detected a pointer year in Douglas-fir that was not found in spruce, specifically 1962, where 100% of the trees on two sites (trees on PO-2 were too young to detect that year) responded with a drop in radial increment (Figure 3).

The following negative pointer years were found on at least two sites for Douglas-fir: 1962, 1993, 2003 and 2012 (Figure 3 and Table 2). In these years, the temperatures in February and March were always below the long-term average, and the winter months were drier with less precipitation in comparison with the long-term average.

On site CE with a more continental climate, we also found 1956 to be a negative pointer year. Since this year was not in our local meteorological archives, we checked on-line sources. It turned out that the winter of 1956 was one of the coldest winters in Europe, with temperatures below −24 °C in Basel, Switzerland and −12 °C in Marseille, France [25,26]. Temperatures in February and March were well below average, which had a negative impact on radial growth in the 1956 growing season. The overall winter situation was very similar, if not more extreme, to that in 1962.

In Douglas-fir, we also found four positive pointer years common to at least two sites: 1989, 1997, 2005 and 2007 (Figure 3 and Table 3). A common feature of the positive pointer years is that the key months for future growth, specifically February and March, were always warmer than average, and the monthly precipitation was within or slightly above the long-term average. None of the positive pointer years were particularly cold in the winter or dry in the summer.

One pointer year was common to both studied tree species: the negative pointer year of 2003. That year was one of the hottest and driest ever recorded in Europe. When such a common pointer year occurs in tree-rings, it is important to determine what the weather conditions were during that year such that the same response was observed in the two physiologically different tree species.

The year 2003 was a markedly negative year for many tree species across Europe, including Douglas-fir and Norway spruce [27]; a drought dragged on until September 2003, when monthly rainfall reached the long-term average for the first time in that year. January and July were slightly below the long-term average, while all other months were very dry, with very little or no precipitation. The small amount of precipitation in July was insufficient to compensate for the lack of precipitation in the previous months. Temperatures in 2003 started with below-average values in January, February and March, but then rose to well above the long-term average between April and August.

In 2003, Douglas-fir was mainly influenced by below-average temperatures in February, the month which has a key effect on the growth of Douglas-fir in the same year. Spruce, however, was negatively affected by a lack of precipitation in the first eight months of the year, including the very dry winter and dry summer months, which are key factors in the radial growth of Norway spruce. Therefore, both tree species responded uniformly with a small radial increment, but for different reasons (Table 2).

## 3. Discussion

### 3.1. Radial Growth of Douglas-Fir and Norway Spruce on Three Sites

On average, the radial growth of Douglas-fir on three locations in Slovenia always surpassed the growth of Norway spruce, regardless of site productivity. On the most productive non-carbonate site, the average radial increment of Douglas-fir was 4.73 mm and that of Norway spruce was 3.48 mm. On the two limestone bedrock sites, Douglas-fir grew better than Norway spruce; however, on site PO-1 there was no major difference between the two species: 3.74 mm for Douglas-fir and 2.40 mm for Norway spruce. On site PO-2, the difference was similar to the most productive site: 4.53 mm vs. 2.17 mm in favour of Douglas-fir.

Studies done on Douglas-fir in provenance trials with as many as eighteen coastal Douglas-fir provenances [13] showed a radial increment comparable to that of Douglas-fir on sites SE of the Alps. Douglas-firs in the provenance trial had a radial increment ranging between 3.47 mm and 4.42 mm. Douglas-fir from Slovenian sites ranked among the better-growing Douglas-firs when compared with those from provenance trials.

Results from the studies in mixed (Scots pine, European larch, Douglas-fir and black pine) even-aged plantations in Switzerland [3] showed that Douglas-fir is performing well in mixed stands, and that its radial increment is superior to that of other tree species. When comparing the radial increment of Douglas-fir from the trial in Switzerland with that from sites in Slovenia, we found that Douglas-fir grew better on sites in Slovenia, with the exception of PO-1, where the radial increment was slightly below that in Switzerland.

In a study by Vitali et al. [28] from the Black Forest, where the radial increment of Douglas-fir was studied in a three-species intermixed Douglas-fir, Norway spruce and silver fir forest, it was determined that Douglas-fir had the highest radial increment of all three studied species. In comparison with Douglas-fir from sites in Slovenia where the combination of Douglas-fir, Norway spruce and silver fir is very common in the close-to-nature forests of the Karst region, we found that the radial increments of Douglas-fir were not as high but very close to those recorded in the Black Forest.

On Czech forest sites, where Douglas-fir was introduced to forest stands at the end of the 19th and in the 20th centuries [17], researchers carried out a study on 18 sites and concluded that the average radial increment was rather small. In the best case it reached 3.66 mm, and in the worst case it was 1.44 mm. These are far below our findings, and it appears that the productivity of the site where Douglas-fir was planted in Czechia is lower than that in Slovenia.

It seems that the radial increments of Douglas-fir are relatively high in regions with predominantly humid climates (e.g., Slovenia, Southern Germany, parts of Switzerland, The Netherlands), and lower in countries with a typical continental climate with colder and drier winters (e.g., Czechia). Our research supports this postulation, given that Douglas-fir is sensitive to very cold (and dry) winter months (typical of a continental climate), which results in a smaller radial increment in the vegetation period that follows this paper and [13,17,28,29].

### 3.2. Climate–Growth Relationship of Douglas-Fir and Comparison to Norway Spruce

Our study shows that the radial increment of Douglas-fir is higher when late winter is mild. On all studied sites, above-average temperature in February is highly correlated with higher radial increment. On two sites, CE and PO-2, above-average temperature in March is also highly correlated with higher radial increment. Precipitation, in general and in any part of the year, is not as important as winter temperatures, although we found some significant correlations with precipitation on sites CE and PO-2 (Figure 2 and Appendix A). However, correlation values for precipitation were lower in comparison with those for temperature. In comparison with Norway spruce, Douglas-fir seems to be more tolerant to above-average summer temperatures, although we assessed that precipitation plays an important role in July, particularly in hot and dry years.

The climate response of Douglas-fir in its natural distribution range in North America (Pacific coast in the USA and Canada) varies across the entire area of distribution. Douglas-fir populations growing in relatively warm and dry climates in Canada have growth patterns correlated mostly with annual precipitation, whereas populations growing in wet and cold climates at high elevations have growth patterns correlated with snowfall, winter and annual temperatures, and ocean–atmosphere climate systems. The strongest response was found in populations growing at climatic extremes [29,30]. A study by Restaino et al. [31] on western Douglas-fir forests in the USA showed that Douglas-fir growth was positively correlated with precipitation and negatively correlated with temperature in the growing season, which is a typical response pattern not only of Douglas-fir but also of many other tree species.

When comparing our results to those from natural Douglas-fir stands in North America, we only found similarities in the climate-growth response with Douglas-fir populations growing in North America’s high-elevation wet and cold climates, where a warmer and more humid late winter-early spring resulted in a better radial increment. Since Douglas-fir seed was imported from North America, we can assume that the seeds for some of the older Douglas-fir populations in Slovenia may have come from such regions in N. America [32].

When comparing the climate–growth response of Douglas-fir in Slovenia with that of planted Douglas-fir in different locations in Europe, we found that the climate–growth response on several sites across Europe was comparable. Vejpustkova et al. [17] came to almost identical conclusions with respect to the radial growth of Douglas-fir in Czechia (NE of our sites). The most critical months for Douglas-fir growth were not the summer months but the winter months of January and February. They also found that precipitation in July and August had a positive effect on radial growth. Our observations regarding the response of Douglas-fir to above-average precipitation in summer months are not as conclusive as those from Czechia. Douglas-fir in Slovenia responds positively to July precipitation; however, this response is not as great as in the case of Czechia. An in-depth analysis of the temporal stability of the precipitation signal in the TRW of Douglas-fir on sites in Slovenia showed that precipitation in July is gaining in importance. This is critical, and suggests that climate is changing in such a way that trees need to adapt to new, potentially drier conditions.

Castaldi et al. [16] investigated the climate–growth relationships of Douglas-fir on two contrasting sites in Italy (W and SW of the sites in Slovenia, and on the southern side of the Alps): a Mediterranean area in southern Italy, and a cooler, moister site in the northern Apennines. They found that temperatures in February and March play a positive role in the growth of Douglas-fir on both sites. At the site in northern Italy, Douglas-fir also responds negatively to late summer temperatures and positively to spring–summer precipitation, which is very similar to our findings. The temporal stability of the climate signal in the two Italian Douglas-fir stands showed that the correlation between radial growth and February temperature is stable over time on the northern site and less so on the southern one. Castaldi et al. [16] also found that precipitation on the northern site is becoming increasingly important. This is highly comparable to the results from Slovenia, which indicate the gradually increasing importance of the July precipitation on the majority of the studied sites (Appendix A). The increasing dependency of Douglas-fir on above-average summer precipitation may possibly have a significant impact on the productivity, and may potentially increase mortality of Douglas-fir in the future.

A study on the climate–growth response of Douglas-fir on 26 forest sites in NW Poland [15], locations north of our sites and north of the Alps, showed that January–March temperature had a statistically significant influence on the growth of Douglas-fir. Principal component analysis revealed that the first component (January–March temperature) explained more than 60% of the entire variability. The second principal component related to precipitation in the summer months accounted for only 8% of the variability. This means that although precipitation plays a role in tree-ring formation, its influence is rather small. This response to precipitation is similar to the response of Douglas-fir to precipitation in Slovenia.

Based on our results and those from different studies across Europe, we can conclude that the climate signal in Douglas-fir tree-rings is comparable over a wide geographical area and diverse climate zones (Figure 4). This indicates that Douglas-fir prefers two main types of climates in Europe: (1) cold to temperate winters, no dry season and (2) warm but not hot summers (types Dfb and CfB after Köppen-Geiger climate classification).

An analysis of the spatial extent of the Douglas-fir climate signal, specifically the response of the TRW to February and March temperatures, showed a very coherent and widespread climate signal. This is quite rare in European tree species, with only a few (e.g., silver fir, larch and partially pedunculate oak) displaying such a widespread climate signal. This climate signal also corresponds well with both of the above-mentioned climate types.

### 3.3. Douglas-Fir Response in Extreme Years

The response of trees in extreme years, and especially in extremely warm and dry years, provides insight into a tree species’ response to extreme climatic conditions and its plasticity. Douglas-fir’s natural area of distribution is characterised by a wide range of climate conditions, parent materials, aspects and slopes, and soil textures and sites [20]. It grows best in deep, moist and well-drained soil at mid-elevations with plenty of rainfall. It has a deep root system which can retrieve soil water from deeper layers of the soil. This is in contrast to Norway spruce, which has a shallower root system and is therefore more sensitive to drought.

Pointer years [33] are associated with favourable or unfavourable growing conditions in a certain year. They are a good representation of the common response of trees in a certain year to a common environmental or climatic driver that causes the formation of a wide or narrow tree-ring. In the context of this research, we only focus on negative pointer years in Douglas-fir and Norway spruce and compare them between species and with pointer years detected in Douglas-fir on the European scale.

Based on the literature and our studies, Douglas-fir has not had many negative pointer years between 1900 and the present. In our study, we found five negative pointer years on two out of three sites (trees on one site were too young to capture during pre-1970 pointer years). All negative pointer years in Slovenia were associated with very cold and dry winter months. Two were also associated with very hot and dry summers (1993 and 2003) (Table 2). Some, but not all, extreme years were European wide (e.g., 1956, 2003 or 2013). The year 2003 seems to be negative pointer year for Douglas-fir across Europe. Several authors report that year as being critical for tree growth in general, and for Douglas-fir specifically [3,13,14,17,28]. The year 1976 was also a very warm and dry year in many parts of Europe, but not in Slovenia. We could not detect this pointer year in any of the studied tree species.

Positive pointer years in the tree rings of Douglas-fir in Slovenia are all associated with a mild winter, especially in February, and with an average amount of precipitation throughout the growing season.

The response of trees in extreme years not only shows how trees respond in extreme years, but also how trees will possibly respond to a warmer and drier climate (as predicted by climate models) in the future [34]. Based on pointer-year analysis, we can conclude that the growth response of Douglas-fir in extreme years mainly depends on the winter temperature (February) and on a sufficient amount of precipitation in the peak summer months (July and August). Taking into account different climate change scenarios for Europe [35], we can anticipate that with increasing temperature and, in the best case, stable precipitation patterns, Douglas-fir will have difficulty maintaining high growth rates at increasing vapour-pressure deficits.

## 4. Materials and Methods

### 4.1. Sampling Locations

Douglas-fir samples were collected at three locations in Slovenia: Postojna 1 (PO-1-DF), Postojna 2 (PO-2-DF) and Celje (CE-DF)—see Figure 5. Research site Postojna 1 is located on a slope with an eastern exposure on pronounced Karst terrain. Research site Postojna 2 is located on Karst terrain on a ridge with shallow soil and northern exposure, and research site Celje is located on deep soil with a northern exposure. The reason for selecting research plots with such diverse stand conditions is to study the response of the same two tree species in different stand conditions.

We also collected Norway spruce samples for reference on three locations with similar ecological characteristics and as close as possible to the Douglas-fir sites: Ravnik (PO-1-NS), Verd (PO-2-NS) and Celje (CE-NS)—see Table 4 and Figure 5.

### 4.2. Sample Collection and Tree-Ring Width Analysis

For the tree-ring width (TRW) analysis, we sampled 20 trees per each location and tree species, which resulted in a total of 60 sampled trees for Douglas-fir and 60 sampled trees for Norway spruce. From each tree we took two 5-millimetre cores, for a total of 240 cores, 120 cores per tree species. All sampled trees were healthy, co-dominant trees with no visible signs of stem damage or any kind of declining tree vitality. Cores were stored in plastic holders and transported to the laboratory.

Once in the laboratory, the cores were dried and then glued onto wooden holders, and sanded with progressively finer sanding paper (up to 800 grit) on an industrial belt sander in order to achieve a highly polished surface and excellent visibility of tree rings. Polished cores were scanned using the ATRICS image capturing system [36]. Core images were then transferred to CooRecorder and CDendro programs (Cybis, Sweden) for measuring and cross-checking. Finally, absolute dating was done in the PAST-5 program (SCIEM, Austria).

Individual TRW series were standardized to remove long-term trends [37] using a 67% cubic smoothing spline with a 50% frequency cut-off in the *dplR* library [38,39] of the R program [40]. Departures of the measured values from the regression curve were calculated as the quotient between the measured tree-ring width and fitted value, resulting in a dimensionless index with a mean of 1. The purpose of this step is to remove factors that are not connected with climate, such as tree age and the effects of stand dynamics [37]. Index values were pre-whitened using an autoregressive model that was selected based on the minimum Akaike criterion, and were combined across all series using a biweight robust estimation of the mean in order to exclude the influence of outliers. The *dplR* produces two types of chronologies: standardized (STD) and residual (RES). In this research, we used the STD chronology, which is a robust estimate of the arithmetic mean and contains autocorrelation [37].

### 4.3. Meteorological Data

We used climate data from two sources: local and gridded climate data. The local data source was too brief for statistical evaluation of the climate–growth relationship; therefore, we used it only as a comparison for the gridded data to see whether the gridded data represent local climate conditions reasonably well.

As a local source of meteorological data, we used the data from two meteorological stations: Postojna (period 1961–2019, N 45° 47′, E 14° 15′, 743 m a.s.l.) and Celje (period 1977–2019, N 46° 12′, E 15° 16′, 688 m a.s.l.). Gridded meteorological data were obtained for two grid cells, one for sites Postojna 1 and 2 (PO-1, PO-2) and one for Celje (CE), from the Climate Research Unit of the University of East Anglia (Norwich, UK) TS 4.03 database (period 1901–2019, spatial resolution 0.5 × 0.5°) [41] (Figure 6).

Comparison of the two data sets showed that the station data indicated a slightly warmer and drier climate compared to the CRU TS 4.03 data set. Summer precipitation minima are larger in the station data set than in the gridded one. Despite some minor differences for the overlapping period, gridded data were used for the analysis presented in the Results section.

The climate of the studied region ranges from transitional sub-Mediterranean-continental to continental. Sites PO-1 and PO-2 have a transitional sub-Mediterranean-continental climate with 1727 mm precipitation per year and an average yearly temperature of 7.9 °C. In this type of climate, there is a high amount of precipitation in the period between September and December (702 mm), with the temperature in this period ranging between 0 °C and 10 °C; January and February are the coldest months, with temperatures below zero and a relatively large amount of snow. According to the long-term average, February is also the driest month (101 mm of precipitation), and January is the coldest month in the entire year (−1.6 °C). A potential lack of precipitation can occur in July and August, although the long-term average shows that this occurs infrequently. The average monthly temperature during the growing season is between 11.7 °C in May and 13.4 °C in September, with a peak in July (17.2 °C). The climate in the PO-1 and PO-2 regions has changed in the last 3 decades. The average monthly temperature has increased to 16.3 °C in June (+1.3 °C), 18.4 °C in July (+1.2 °C) and 18.1 °C in August (+1.4 °C). The precipitation regime has also changed. Late autumn and early winter peaks are more pronounced, with an increase of between 15 and 25 mm/month (Figure 6).

Site CE has a continental climate with 1298 mm of precipitation per year and an average yearly temperature of 8.7 °C. On average, 660 mm of precipitation is recorded during the growing season (May–September). The average yearly temperature of the coldest month, January, is −1.4 °C, and the hottest month, July, is 18.2 °C. June is the wettest month, with 146 mm of precipitation. Drought may occur in July and August. The climate in the CE region has changed in the last three decades as well. The average monthly temperature in all months has increased and the monthly sum of precipitation has decreased. In general, the average yearly temperature is 1 °C higher and the yearly amount of precipitation is 58 mm lower than in previous decades (Figure 6).

### 4.4. Analysis of the Climate–Growth Relationship

After de-trending, TRW chronologies were compared to average monthly temperatures and to the monthly sum of precipitation using a bootstrapped correlation coefficient calculation in the *treeclim* library [42] of the R program [40]. Several combinations of monthly temperature and precipitation data were tested against the tree-ring widths of each studied tree species, in order to find the best possible combination of influential climate variables. A 25-year window with a one-year overlap for the calculation of the bootstrapped correlation between monthly temperature and precipitation and tree-ring widths of both studied species was used to assess the temporal stability of the climate–growth relationship.

Pointer year analysis was done for each tree species on all three locations. We used standard criteria for pointer year selection, as described in Schweingruber et al. [33]. A year was recognized as a pointer year when 80% of at least 13 trees per site and species responded with an increase or decrease in tree-ring width in comparison to the prior year.

In the analysis, we used monthly gridded temperature and precipitation data (0.5 × 0.5° grid) from the CRU TS database [41], available at the KNMI Climate Explorer website [43,44]. Statistical analysis was done in R libraries *dplR* [38] and *treeclim* [42], and graphs were created using IgorPRO.

## 5. Conclusions

Based on our research and comparisons with other studies across Europe and North America, we can conclude that:Douglas-fir is more drought tolerant than Norway spruce, and as such is better adapted to increasing temperatures and more frequent occurrences of drought events in Slovenia. In part, this relates to its deeper root system than that of Norway spruce, and hence better accessibility to deeper lying water.The positive response of Douglas-fir to warmer and wetter winter months is beneficial, as winters are not as cold as they used to be. However, the combination of cold and dry winters and hot and dry summers have negative effects on Douglas-fir radial growth. These effects are similar to the effects of a hot and dry summer on Norway spruce radial growth. Both tree species respond in the same way with a significant decrease in radial increment.Douglas-fir is not very sensitive to lack of precipitation in the summer months, but temporal analysis of the correlation between tree- ring widths and summer precipitation at sites in Slovenia shows an increasing importance of summer precipitation (especially precipitation in June and July), suggesting that precipitation may become a growth-limiting factor for Douglas-fir in the future.The positive response in radial growth of Douglas-fir to warmer and wetter winter months is not limited to sites in Slovenia; its spatial outreach is much wider, extending throughout western and central Europe as well as in the northern parts of the Balkan and Apennine Peninsulas.From the climate–growth point of view, it seems that Douglas-fir can be a good substitute for Norway spruce in part of the current mixed forest stands in Slovenia; however, this is not only a climate–growth related issue. The successful introduction of Douglas-fir into Slovenian, close-to-nature managed forests is also a forest management and legislative problem. As a potentially invasive alien species, Douglas-fir is not allowed to be planted in Slovenian forests, and knowledge about Douglas-fir tending and management is still limited.Responses in extremely dry years (e.g., 2003) have shown that Douglas-fir can survive shorter dry periods on drought-prone sites, such as the High Karst in Slovenia (permeable limestone bedrock, shallow soil), but in the long term it is not advisable to plant Douglas-fir on drought-prone sites, especially considering current climate change.

## Figures and Tables

**Figure 1 plants-11-01571-f001:**
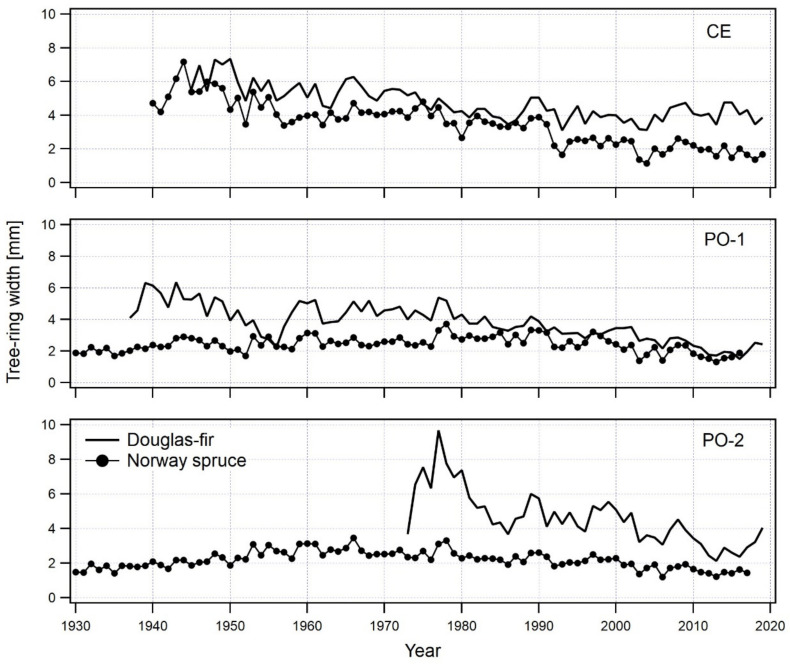
Comparison of tree-ring width chronologies for Douglas-fir and Norway spruce per site. Norway spruce chronologies on PO-1 and PO-2 sites are two years shorter since the samples were collected two years earlier.

**Figure 2 plants-11-01571-f002:**
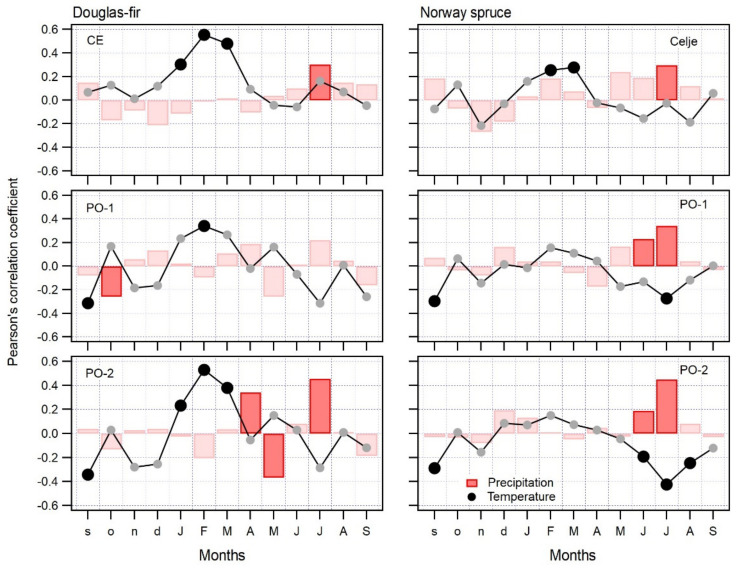
Climate–growth response of Douglas-fir (**left** side panel) and Norway spruce (**right** side panel).

**Figure 3 plants-11-01571-f003:**
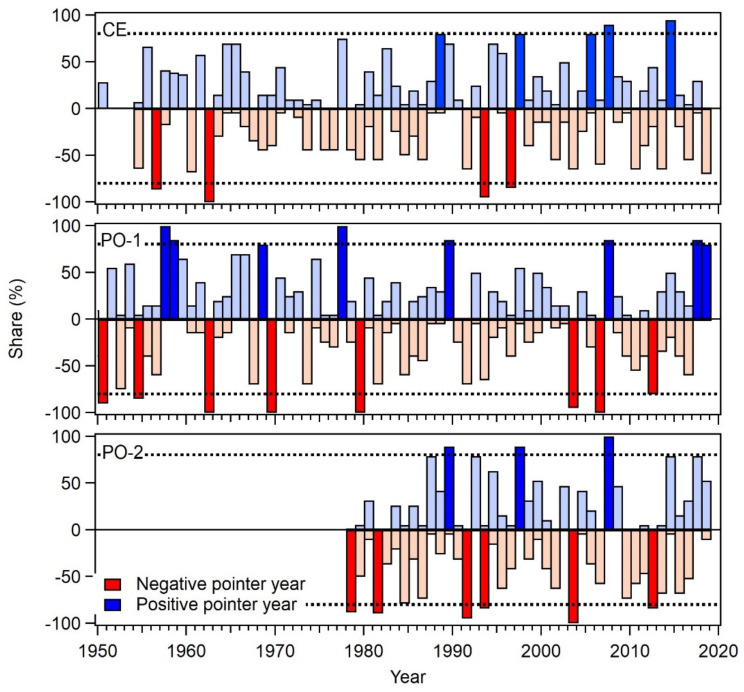
Positive and negative pointer years for Douglas-fir on all three studied sites. The horizontal dashed line represents the 80% threshold for a year to be qualified as a positive or negative pointer year. Emphasised colours indicate significant pointer years with more than 80% of at least 13 trees in a particular year showing an increase or decrease in tree-ring width compared to the previous year.

**Figure 4 plants-11-01571-f004:**
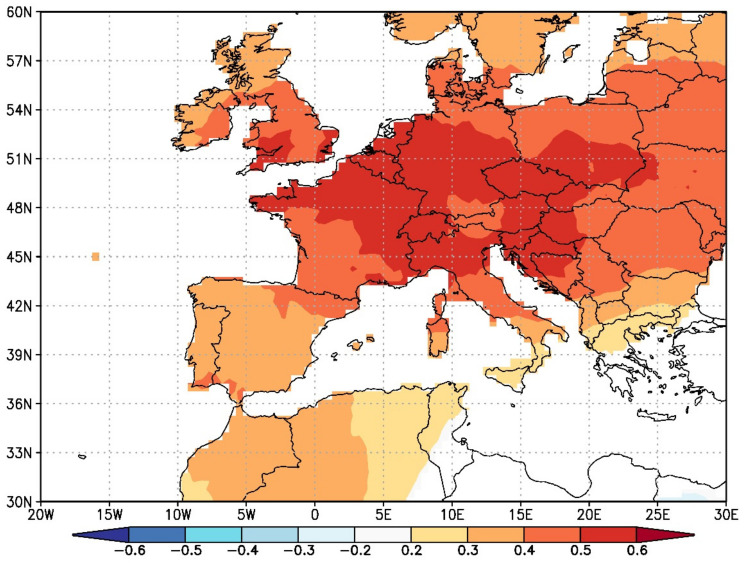
Spatial point correlation between averaged February and March temperatures (1937–2019) and combined Douglas-fir tree-ring chronology for Slovenia. The map shows the spatial extent of Douglas-fir climate–growth signal from Slovenia.

**Figure 5 plants-11-01571-f005:**
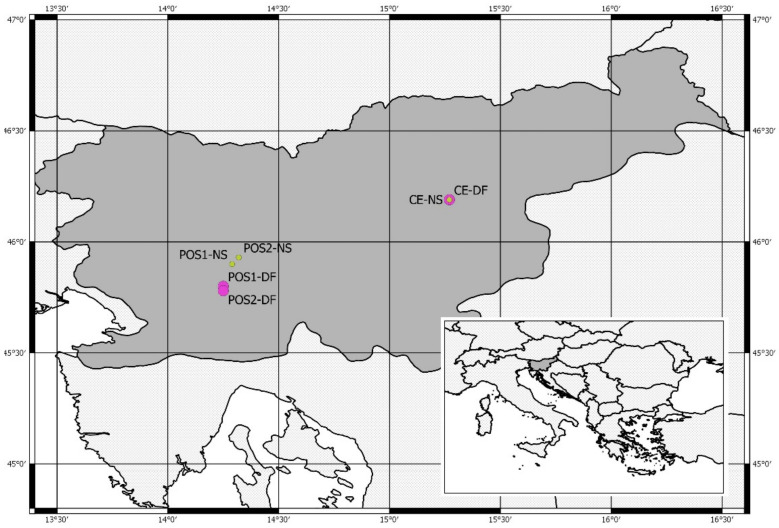
Sampling locations of Douglas-fir (DF) and Norway spruce (NS) in Slovenia (large map) and the location of Slovenia in the wider regional context (smaller map, bottom right).

**Figure 6 plants-11-01571-f006:**
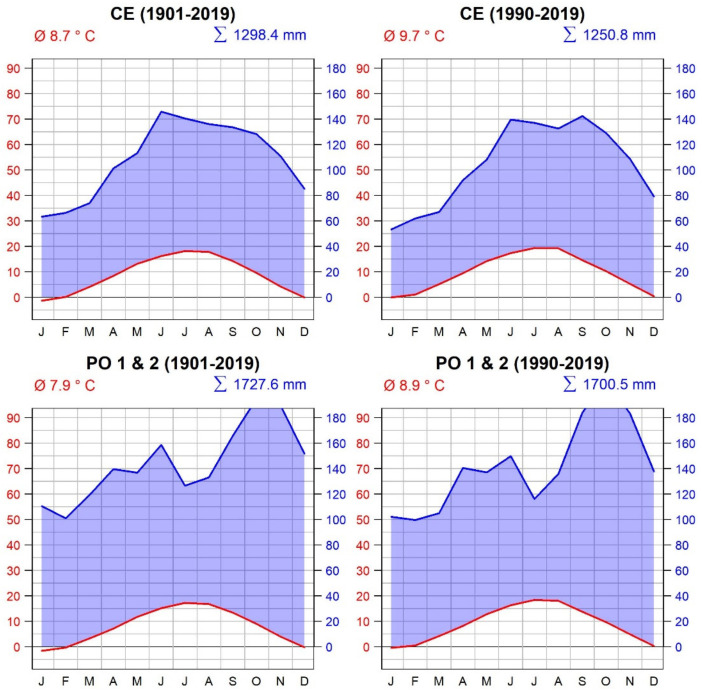
Climate diagrams for the sampling locations CE (**top**) and PO-1 and PO-2 (**bottom**). Sampling locations PO-1 and PO-2 are close to each other; therefore, the same climate data set is used. Climate diagrams are based on CRU TS 4.03 gridded climate data. The **left** panel (**top** and **bottom**) shows the entire period of available meteorological data (1901–2020), while the **right** panel (**top** and **bottom**) shows only the last 30 years (1990–2020) for all three sites.

**Table 1 plants-11-01571-t001:** Basic data on the analysed Douglas-fir and Norway spruce trees on all three sites (PO-1, PO-2 and CE). TRW—tree-ring width, DBH—diameter at breast height measured with a diameter tape, Std. dev.—standard deviation.

Location	Tree Species	Age (Years)	DBH (cm)	Std. Dev. DBH (cm)	TRW * (mm)	Std. Dev. TRW (mm)
CE	Douglas-fir	70	69.60	9.51	4.73	0.986
	Norway spruce	71	52.20	4.89	3.48	1.29
PO-1	Douglas-fir	79	75.56	9.39	3.74	1.34
	Norway spruce	96	57.06	7.53	2.40	1.03
PO-2	Douglas-fir	45	49.33	5.87	4.53	1.79
	Norway spruce	121	59.84	11.66	2.17	0.75

* Tree-ring width is the average of the tree-ring width measurements from two cores per tree.

**Table 2 plants-11-01571-t002:** Common negative pointer years for Douglas-fir on sites PO-1, PO-2 and CE (source of meteorological data: https://meteo.arso.gov.si/met/sl/climate/current/last-12-months/archive/ accessed on 3 February 2022).

Year	Temperature	Precipitation	Sites
*Common to all three sites*			
	No common negative pointer years		
*Common to two sites*			
1956	February and part of March extremely cold across Europe	No data available	PO-1 and CE, trees on PO-2 too young
1962	Very cold March, close to the long-term record low, below-average temperatures in the first half of the year (including July)	Above-average amount of precipitation between January and July	PO-1 and CE, trees on PO-2 too young
1993	February and March very cold, close to the long-term low, other months close to the long-term average	Below-average amount of precipitation between January and July, absence of snow	PO-2 and CE
2003	February very cold, March within the long-term average, May–August significantly above the long-term average	Entire year very dry, all but late autumn months below the long-term average	PO-1 and PO-2
2012	February very cold, March very warm, June–September notably above the long-term average	January–March very dry, April–July within the long-term average, August dry	PO-1 and PO-2

**Table 3 plants-11-01571-t003:** Common positive pointer years for Douglas-fir on sites PO-1, PO-2 and CE (source of meteorological data: https://meteo.arso.gov.si/met/sl/climate/current/last-12-months/archive/ accessed 3 February 2022).

Year	Temperature	Precipitation	Sites
*Common to three sites*			
2007	Slightly above-average temperature between January and July, then within the long-term average	Precipitation in the entire year within the long-term average with the exception of a very dry April	PO-1, PO-2 and CE
*Common to two sites*			
1989	February and March temperatures above average, summer temperatures below average	April with above-average amount of precipitation, June–July average and August close to the long-term maximum of precipitation for August	PO-1 and PO-2
1997	February and March temperatures above average	Amount of precipitation in the period February–April above average, later within the long-term average	PO-2 and CE

**Table 4 plants-11-01571-t004:** Site characteristics and sample replication for Douglas-fir and Norway spruce sample pool.

Douglas-fir			
	*Postojna-1*	*Postojna-2*	*Celje*
Local site name	Mačkovc	Golobičevec	Pečovnik
	PO-1-DF	PO-2-DF	CE-DF
Coordinates	N 42.57°, E 20.03°	N 42.63°, E 19.85°	N 46.19°, E 15.27°
Elevation	584–682 m	670–790 m	465–650 m
Slope	16°	18°	25°
Exposition	E	N	N
Soil type	Brown soil on limestone	Shallow brown soil on limestone	Deep brown soils on silicate
Number of cores for tree-ring analysis	40 cores (20 trees) each plot		
Norway spruce			
Local site name	Ravnik	Verd	Pečovnik
	PO-1-NS	PO-2-NS	CE-NS
Coordinates	N 45.90°, E 14.29°	N 45.93°, E 14.32°	N 46.19°, E 15.27°
Elevation	655–795 m	535–700 m	465–650 m
Slope	20°	25°	25°
Exposition	SW	SE	N
Soil type	Brown soil on limestone	Shallow brown soil on limestone	Deep brown soils on silicate
Number of cores for tree-ring analysis	40 cores (20 trees) each plot		

## Data Availability

Tree-ring width data for Douglas-fir and Norway spruce are available on the Mendeley Data repository under Levanic, Tom (2022), “Douglas-fir and Norway spruce tree-ring data from three sites in Slovenia”, Mendeley Data, V1, https://doi.org/10.17632/32zydvmzjd.1 (accessed on 3 February 2022).

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
