# Peer review of "Effects of Climate on Douglas-fir (Pseudotsuga menziesii (Mirb.) Franco) Growth Southeast of the European Alps"

_plants, 2022, doi:10.3390/plants11121571_

Round 1
Reviewer 1 Report
The objective of the manuscript was to research and compare radial growth between Norway spruce and Douglas fir on three different sites in Slovenia, and also to investigate responses of these tree species to climate factors and sensitivity to extreme climate events. In a context of climate change, this study has high relevance on local level and also worldwide. The study objectives were supported by the reasonable field sampling and methods of data analyses. Results and methods chapters are clearly presented in the paper. The obtained results indicated higher resistance of Douglas fir to extreme drought as a potential substitute of Norway spruce in expected climate impacts. In discussion section, the obtained results were discussed in accordance with relevant studies in Europe and North America, and accordingly, this paper may give research contribution of the topics.
Although some conclusions are visible in discussion section, the conclusions sections can be added in the paper in order to highlight main conclusions of the study. Minor edits and checks of text are needed.
Other small corrections and comments are included within text of the MS.
Overall, I can recommend publishing this manuscript in the Plants journal after minor revision.

Author Response
R-1
The objective of the manuscript was to research and compare radial growth between Norway spruce and Douglas fir on three different sites in Slovenia, and also to investigate responses of these tree species to climate factors and sensitivity to extreme climate events. In a context of climate change, this study has high relevance on local level and also worldwide. The study objectives were supported by the reasonable field sampling and methods of data analyses. Results and methods chapters are clearly presented in the paper. The obtained results indicated higher resistance of Douglas fir to extreme drought as a potential substitute of Norway spruce in expected climate impacts. In discussion section, the obtained results were discussed in accordance with relevant studies in Europe and North America, and accordingly, this paper may give research contribution of the topics.
Although some conclusions are visible in discussion section, the conclusions sections can be added in the paper in order to highlight main conclusions of the study. Minor edits and checks of text are needed.
- Really brief description of the used model was added. We think that for the introduction it is enough to explain that we used empirically based random forest model. Model itself has no name and it was developed “in house” for the purpose of the project dealing with the future of Norway spruce stand in Slovenia.
- One sentence explaining moving window correlations was rewritten to improve readability.
- Conclusions added after the Material and Method chapter
Other small corrections and comments are included within text of the MS.
- All suggestions for corrections were accepted (also those in the PDF file)

Reviewer 2 Report
Dear Authors,
The subject of the study is interesting and topical, with scientific and practical importance.
The introduction is presented correctly, in accordance with the subject. Numerous scientific articles, in concordance to the topic of the study, were consulted.
Methodology of the study was clearly presented, and appropriate to the proposed objectives.
The obtained results are important and have been analyzed and interpreted correctly, in accordance with the current methodology. It is recommended that some issues be reviewed.
The discussions are appropriate, in the context of the results, and was conducted compared to other studies in the field.
The scientific literature, to which the reporting was made, is recent and representative in the field.
Some suggestions and corrections were made in the article.
The following aspects are brought to the attention of the authors.
1.
Tables settings
It is recommended to check the tables settings in relation to Instructions for Authors, and Microsoft Word template, Plants journal
Eg
Page 3, Table 1.
Page 9, Table 2
Styles:
MDPI_4.1_table_caption
MDPI_4.2_table_body
The inner lines are recommended to be removed
2.
Figures in the content of the article
According to Instructions for Authors, and Microsoft Word template, “Figures should be placed in the main text near to the first time they are cited.”
It is recommended to check if this is possible
3.
References
It is recommended to revise the References chapter, and some corrections, as appropriate.
Abbreviated Journal Name
eg
Page 18, row 611
"For. Ecol. Manag." instead of “Forest Ecology and Management”
Styles: MDPI_7.1_References
Some suggestions have been made in the article, References chapter

Author Response
R-2
Dear Authors,
The subject of the study is interesting and topical, with scientific and practical importance.
The introduction is presented correctly, in accordance with the subject. Numerous scientific articles, in concordance to the topic of the study, were consulted.
Methodology of the study was clearly presented, and appropriate to the proposed objectives.
The obtained results are important and have been analyzed and interpreted correctly, in accordance with the current methodology. It is recommended that some issues be reviewed.
The discussions are appropriate, in the context of the results, and was conducted compared to other studies in the field.
The scientific literature, to which the reporting was made, is recent and representative in the field.
Some suggestions and corrections were made in the article.
- All correction were considered and accepted (including those in PDF)
Tables settings
It is recommended to check the tables settings in relation to Instructions for Authors, and Microsoft Word template, Plants journal
Eg
Page 3, Table 1
Page 9, Table 2
Styles:
MDPI_4.1_table_caption
MDPI_4.2_table_body
The inner lines are recommended to be removed
- All suggested corrections have been done
2.
Figures in the content of the article
According to Instructions for Authors, and Microsoft Word template, “Figures should be placed in the main text near to the first time they are cited.”
It is recommended to check if this is possible
- Checked, figures are as close as possible.
3.
References
It is recommended to revise the References chapter, and some corrections, as appropriate.
Abbreviated Journal Name
eg
Page 18, row 611
"For. Ecol. Manag." instead of “Forest Ecology and Management”
Styles: MDPI_7.1_References
- Done and corrected. However, MDPI reference style was always updated with some internal Endnote style, so we try to come as close to MDPI style as possible.
Some suggestions have been made in the article, References chapter
- Suggestions in the attached PDF were inspected and corrections applied
